# NOx Photooxidation over Different Noble Metals Modified TiO2

**Kinga Skalska [1,2,\*]**, **Anna Malankowska [3]**, **Jacek Balcerzak [2]**, **Maria Gazda [4]**, **Grzegorz Nowaczyk [5]**,
**Stefan Jurga [5,†]** and **Adriana Zaleska-Medynska [3]**

[1] Department of Separation Science, LUT University, Sammonkatu 12, 50130 Mikkeli, Finland
[2] Department of Molecular Engineering, Faculty of Process and Environmental Engineering, Lodz University of Technology, Wolczanska 213, 90-924 Lodz, Poland; jacek.balcerzak@p.lodz.pl
[3] Department of Environmental Technology, Faculty of Chemistry, University of Gdansk, Wita Stwosza 63, 80-308 Gdansk, Poland; anna.malankowska@ug.edu.pl (A.M.); adriana.zaleska-medynska@ug.edu.pl (A.Z.-M.)
[4] Department of Solid State Physics, Faculty of Applied Physics and Mathematics, Gdansk University of Technology, 80-952 Gdansk, Poland; margazda@pg.edu.pl
[5] NanoBioMedical Centre, Adam Mickiewicz University, Wszechnicy Piastowskiej 3, 61-614 Poznan, Poland; nowag@amu.edu.pl
\* Correspondence: kinga.skalska@lut.fi
† Deceased.

**Abstract:** We compared the activity enhancement effect of noble metal deposited on TiO2 in photocatalytic nitrogen oxides oxidation. Titanium dioxide was decorated with Ag, Au, Pt or Pd in the sol-gel process. Synthesized catalysts were characterized by X-ray diffraction (XRD), Brunauer–Emmett–Teller measurement (BET), X-ray photoelectron spectroscopy (XPS), transmission electron microscopy (TEM) and energy dispersive X-ray analysis (EDX). All catalysts together with pure TiO2 obtained by sol-gel (SG) technique were tested for their photocatalytic activity towards nitrogen oxide oxidation (high concentrations of 50, 150 and 250 ppm). FTIR spectrometry was used to determine the gas phase composition and identify TiO2 surface species. The Ag0.1 sample turned out to be deactivated within 60 min of UV/Vis irradiation. Photocatalytic oxidation rate towards NO2 turned to be the highest over SG (photocatalyst without metal deposition). NO2 formation was also observed for Au0.1, Au0.5, Pt0.1, Pt0.5 and Pd0.1. The best NOx removal, i.e., conversion to final product HNO3 was obtained with the Au0.5 photocatalyst.

**Keywords:** NOx emission abatement; TiO2; surface decoration; noble metals; nitrogen oxide; photocatalysis

## 1. Introduction

Ongoing deterioration of air quality endangers human health and degrades our environment. There are many pollutants that contribute to the seriousness of the problem, i.e., sulphur dioxide, nitrogen oxides, particulate matter, benzo(a)piren, hydrocarbons, etc. Emissions of nitrogen oxides (NOx = NO and NO2) into the atmosphere lead to the occurrence of tropospheric ozone and urban photochemical smog as well as acid rains. Various methods exist to reduce NOx emission from industrial sources. However, in the face of aggravation of emission limits we need more effective and green solutions. Law regulations are a tool to force the industrial sector and the society to pay more attention to the problem of ongoing air quality deterioration. Recently Verbruggen, (2015) pointed to the statistical data from the European Environment Agency (EEA) in which it is stated that human life expectancy is reduced by more than eight months on average and in the most polluted cities even by more than two years [1].

Some of the new approaches towards the problem of NOx pollution have been described by Lasek et al., 2013 and Nguyen et al., 2020 in their review papers [2,3]. Wet processes are economic and efficient way to remove NO2 from industrial exhaust gases, unfortunately they cannot remove NO because of its low solubility in water. It is well

known fact that nitrogen monoxide is not soluble in water, and to increase the effectiveness of the $NO_x$ removal process it is necessary to raise the amount of nitrogen dioxide in the total $NO_x$ stream. Nitric oxide can be easily oxidized into $NO_2$ through chemical reactions with various oxidizing agents (e.g., ozone) [4]. However, generation of ozone is connected with high energy demand. A new idea is to employ photocatalysis to increase the $NO_2/NO_x$ ratio in industrial flue gases. To the best of our knowledge the number of publications in this area is scarce. Still many questions remain unaddressed, first of all is photocatalytic oxidation (PCO) a feasible solution, then which of the photocatalytic materials are preferable and finally what the influence is of process parameters on the effectiveness of NO photocatalytic oxidation into $NO_2$ and $HNO_3$. Some of these queries were addressed in the current work.

Possible applications of $TiO_2$ and other nanomaterials are vast, i.e., photocatalytic water splitting, photocatalytic water and air purification, photovoltaics etc. [5–7]. It has been already proven that photocatalytic nanomaterials need to be tailored for a specific application and in the case of environmental applications for a specific target-pollutant. Nevertheless, studies aimed at enhancing photocatalytic activity of such materials are still often focused on presenting photocatalytic activity in respect to a model pollutant in the aqueous solution. Most commonly, this model pollutant is phenol, methylene blue, rhodamine B or chloroform etc. [8,9]. Studies of nanomaterials in the gas phase are scarcer, with studies focusing on elimination of volatile organic compounds (VOC) and inorganic gases [6]. One of the proposed applications of novel nanomaterials in air purification is the removal of nitrogen oxides.

Ag, Au, Pt or Pd nanoparticles can promote both activity of, e.g., $TiO_2$ under UV and Vis light, through restraining electron-hole recombination or appearance of surface plasmon resonance (SPR). Bowering et al. [10] and Sofianou et al. [11] showed Ag decorated $TiO_2$ with enhanced activity in UV light in the NO photocatalytic removal. Similar effects were separately reported for Au [12], Pd [13] and Pt [14] decorated $TiO_2$ based materials. Lately Hernández-Rodríques et al. [15] compared gold and platinum decorated $TiO_2$ for 500 ppb NO photocatalytic oxidation under UV and Vis irradiation. Best results were obtained for P25 with Au nanoparticles decorated by chemical reduction method.

The fundamental mechanism of the photocatalytic NO oxidation process as well as the spectrum and the fate of reaction products are still poorly understood [10] especially for processes carried out at high NO initial concentrations (around hundreds ppm).

In the present study we have juxtaposed titanium dioxide decorated separately with four noble metals (Ag, Au, Pt, Pd) for their efficiency in the photocatalytic nitrogen oxides oxidation. To the best of our knowledge no one has previously compared anatase $TiO_2$ decorated with those noble metals for its activity in photocatalytic nitrogen oxides oxidation. It is worth noting that FTIR gas phase measurements were used for the first time to determine reaction products in $NO_x$ photocatalytic oxidation.

## 2. Results and Discussion

### 2.1. Catalyst Characterization

XRD patterns obtained for metal decorated photocatalysts are presented in Figure S1. Typical diffraction reflections corresponding to anatase phase ($2\theta = 25°, 48°, 55°$) were observed in all the synthesized photocatalysts (Figure S1). Additionally, one weak reflection at $2\theta = 31°$ corresponding to the brookite phase was observed for all the samples. For the sample Au0.5 the reflections of metallic gold were observed at ($2\theta = 38.2°, 44.4°$). Crystallite size was estimated using the Scherrer analysis of the (101), (004) and (200) XRD reflections. Instrumental width was determined based on the pattern of $LaB_6$. The crystallites are rather small, with sizes between 5 and 10 nm. The sizes of crystallites seem to be isotropic with no strong dependence on the crystallographic direction. BET surface area, and pore size were gathered in Table S2. BET surface area varied from 101.5 to 142.8 $m^2$ $g^{-1}$. The pore size of the prepared materials was between 0.05 and 0.07 $cm^3$ $g^{-1}$.

Atomic composition and chemical state of elements imbedded in the surface layer (5–10 nm) of the synthesized photocatalysts was investigated by XPS analysis. Relative atomic composition is presented in Table 1. Chemical state analysis of elements was performed based on the obtained narrow scans (Figure 1).

**Table 1.** Surface composition of studied photocatalytic films.

| Element | Band | Ag 0.1 | Ag 0.5 | Au 0.1 | Au 0.5 | Pt 0.1 | Pt 0.5 | Pd 0.1 | Pd 0.5nr | Pd 0.5r |
|---|---|---|---|---|---|---|---|---|---|---|
| | | | | | Content (% at.) | | | | | |
| Titanium | Ti 2p | 22.61 ± 0.14 | 24.04 ± 0.24 | 23.48 ± 0.23 | 22.17 ± 1.45 | 23.39 ± 0.18 | 23.91 ± 0.28 | 24.65 ± 0.24 | 24.37 ± 0.16 | 24.10 ± 0.13 |
| Oxygen | O 1s | 53.80 ± 0.49 | 55.74 ± 0.17 | 56.87 ± 0.04 | 55.05 ± 1.65 | 57.17 ± 1.39 | 57.43 ± 0.50 | 58.64 ± 0.70 | 57.51 ± 0.47 | 56.50 ± 0.01 |
| Carbon | C 1s | 23.56 ± 0.36 | 19.97 ± 0.08 | 19.62 ± 0.26 | 22.74 ± 3.09 | 19.38 ± 1.56 | 18.49 ± 0.78 | 16.68 ± 0.45 | 17.94 ± 0.33 | 19.24 ± 0.11 |
| Metal band | | Ag 3d | | Au 4f | | Pt 4f | | Pd 3d | | |
| | | 0.04 ± 0.00 | 0.25 ± 0.02 | 0.04 ± 0.01 | 0.06 ± 0.01 | 0.07 ± 0.01 | 0.18 ± 0.01 | 0.03 ± 0.02 | 0.18 ± 0.01 | 0.17 ± 0.01 |

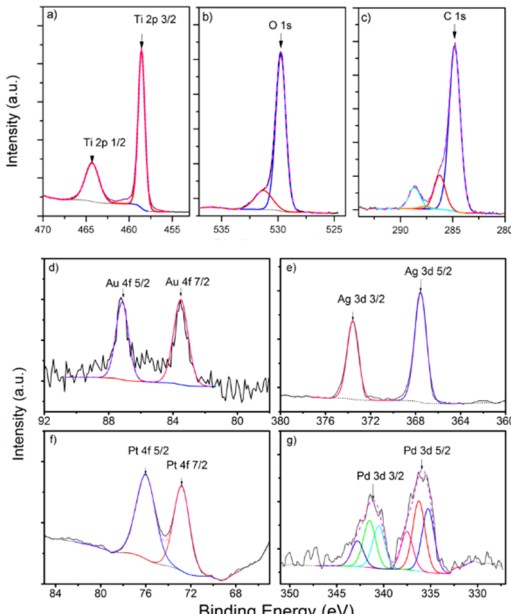

**Figure 1.** XPS high resolution scans for selected samples: (**a**) Ti 2p peaks for sample Ag 0.1, (**b**) O 1s peaks for sample Ag0.1, (**c**) C 1s peaks for sample Ag0.1, (**d**) Au 4f peaks for sample Au0.5, (**e**) Ag 3d peaks for sample Ag0.5, (**f**) Pt 4f peaks for Pt0.5 sample, (**g**) Pd 3d peaks for Pd0.5nr sample.

In respect to the main components, i.e., oxygen, titanium and carbon, the atomic composition of all samples was almost identical. The Ti spectrum (Figure 1a) shows two well resolved peaks. Ti 2p 3/2 at 458.6 eV and 2p 1/2 at 463.9 eV indicate that $Ti^{4+}$ was the dominant surface state of titanium in all the samples. In the oxygen O 1s spectrum two components are present (Figure 1b) at 529.7 eV and 531.3 eV. They were assigned to $TiO_2$ and oxygen in hydroxyl or carbonyl groups bonded with an adventitious carbon, respectively. For samples Pt0.5 and Au0.5 the peak corresponding to the presence of C-O bonds was additionally observed at 532.6 eV. C 1s line was detected, revealing carbonaceous species (Figure 2c) were detected, i.e., C-C/C-H, C-OH, C=O and COOH, at 284.8, 286.3, 287.6 and 288.7 eV, respectively.

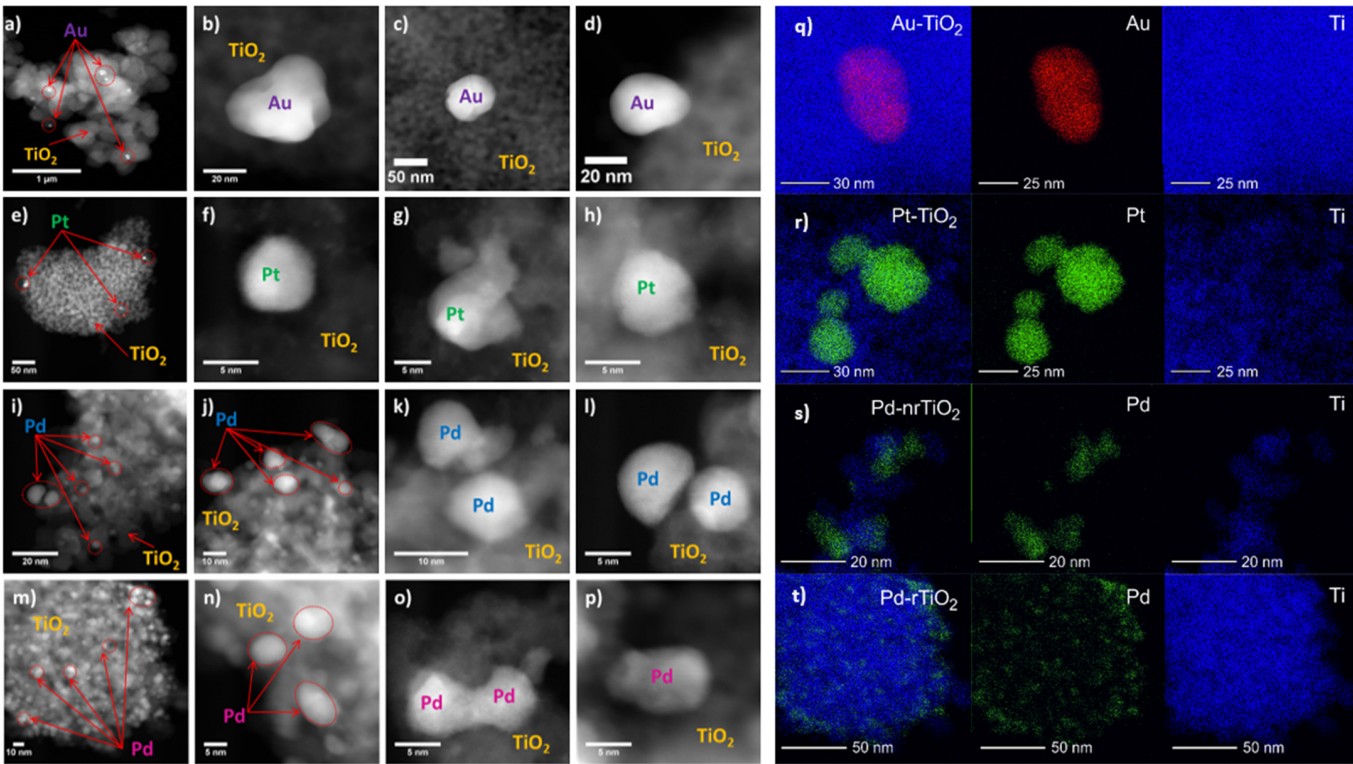

**Figure 2.** Left: HAADF images with *z*-contrast of samples (**a**–**d**) Au0.5, (**e**–**h**) Pt0.5, (**i**–**l**) Pd0.5nr, (**m**–**p**) Pd0.5r, Right: EDS mapping of samples (**q**) Au 0.5, (**r**) Pt0.5, (**s**) Pd0.5nr, (**t**) Pd0.5r (blue is Ti, red is Au and green Pt and Pd).

Noble metals content was also assessed and presented in Table 1. For samples Ag0.1, Au0.1, Pd0.1 the atomic concentration was close to detection limit (0.01% at.), hence it was impossible to perform an analysis of silver and gold chemical states in these samples. Detailed analysis of noble metals chemical states was performed for samples with higher content of noble metals. Figure 1d shows Au doublet Au4f 7/2 and Au4f 5/2 at 83.5 and 87.2 eV, which confirms the presence of metallic gold [15]. In Ag0.5 sample, silver spectrum composed of (Figure 1e) Ag 3d 5/2 and Ag3d 3/2 at 367.5 and 373.5 eV corresponds to the presence of an oxidized form of silver, most likely in $Ag_2O$ [16]. Pt containing sample Pt0.5 (Figure 1f) revealed doublet Pt 4f 7/2 and Pt 4f 5/2 at 72.8 eV and 76 eV, respectively. These peaks were assigned to $Pt^{2+}$, which might be platinum bonded to OH groups [17]. Chemical states of palladium were identified in Pd0.5r and Pd0.5nr samples (Figure 1g). Pd 3d multiple structure was assigned to $Pd^0$ (Pd 3d 5/2 at 335.0 eV), $Pd^{2+}$ as PdO (Pd 3d 5/2 at 336.0 eV) and presumably $Pd^{4+}$ in $PdO_2$ (Pd 3d 5/2 at 337.4 eV) for both Pd reduced and non-reduced samples [13,17]. Variation in content of those components was registered, the content of $Pd^0$ was higher in the reduced catalyst c.a. 46% at. in contrast to 37% at. at the non-reduced catalyst. As can be expected the amount of $Pd^{2+}$ oxidized palladium was lower in the Pd0.5r surface layer when compared with Pd0.5nr, i.e., around 41% at. and 31% at., respectively. The amount of $Pd^{4+}$ form was almost identical in both samples (c.a. 22% at.).

The morphology of the Au, Pt, Pd decorated samples is presented in Figure 2. Metal nanoparticles are seen on STEM images as bright contrasts on the $TiO_2$ surface. For Au0.5 sample, three main fractions of gold nanoparticles were distinguished, i.e., particles with anisotropic shapes (36 to 96 nm), spherical particles (25 to 72 nm) and longitudinal particles (30 to 95 nm). Analysis of Pt0.5 sample revealed spherical particles (7 to 12 nm). $TiO_2$ decorated with Pd had mostly spherical (1 to 10 nm) and longitudinal particles (7 to 40 nm). Nanoparticles obtained on Pd0.5r were generally smaller than in the case of sample Pd0.5nr (i.e., no reducing agent).

The distribution of elements in TiO$_2$ samples decorated with noble metals was assessed using STEM/EDS mapping mode. The red spots were assigned to gold, green spots both platinum and palladium, whereas titanium is marked in blue. It can be observed that Pt and Pd are well distributed on the TiO$_2$ surface, whereas the distribution of Au is less uniform. This was additionally confirmed by EDX analysis of Au0.5 sample (Figure S2). As it can be seen a small amount of gold was detected in the volume of Au0.5 (mean value 0.05% at.). The standard deviation for gold samples was very high, this proves the uneven distribution of Au nanoparticles in the catalyst volume.

### 2.2. Photocatalytic Study

The analysis of the gaseous products of the NO photocatalytic conversion revealed that for all studied materials at least three gaseous products can be detected (NO$_2$, HNO$_2$, HNO$_3$) during irradiation. The FTIR spectra obtained for the photocatalytic process carried out over synthesized catalysts are shown in Figure 3. Analysis of the FTIR spectra proved that the main components are NO (substrate) and NO$_2$ at 1907 cm$^{-1}$ and 1628 cm$^{-1}$, respectively. In addition, four weak peaks were present at 1745, 1540, 1350 and 1316 cm$^{-1}$. Peaks at 1745 cm$^{-1}$ and 1316 cm$^{-1}$ were attributed to NO$_3^-$, whereas the peaks at 1540 and 1350 cm$^{-1}$ were connected with the presence of asymmetrical stretching and symmetrical stretching vibrations in NO$_2^-$ groups. Based on the 1316 cm$^{-1}$ peak height and cross section determined by Wängberg et al. [18] we were able to assess concentrations of gaseous nitric acid, which for all tested materials were in the range 10$^{-8}$ and 10$^{-9}$ mol·L$^{-1}$ (less than 0.5 ppm).

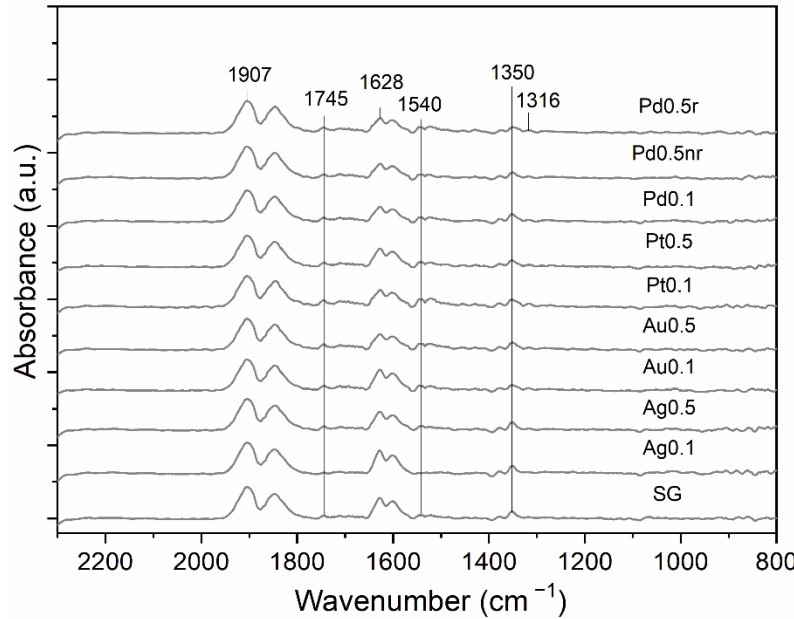

**Figure 3.** FTIR spectra of gas mixture obtained at 60th min of photocatalysis with NO initial concentration equal to 150 ppm.

First of all, the formation of NO$_2$ during the irradiation process is presented in Figure S3. The conversion of NO into NO$_2$ ($\alpha_{\text{NOtoNO2}}$) was calculated accordingly:

$$\alpha_{\text{NOtoNO2}} = \frac{([NO_2]_{\text{out}} - [NO_2]_{\text{in}})}{[NO]_{\text{in}}} \times 100\% \qquad (1)$$

where:

[NO]$_i$—molar concentration of nitrogen oxide at the inlet i-in or outlet i-out of the reactor.

Results for three selected catalysts are presented, i.e., SG, Ag0.1 and Au0.1. As it can be seen the initial conversion over silver decorated material was higher than for SG and Au0.1.

The conversion drops fast to 0 and no steady state analysis could have been performed. We suspect that the loss of activity of Ag0.1 is caused by accumulation of the final reaction product $HNO_3$ which can result in blockage of catalysts' active centers Presence of nitrates at the surface of catalysts after reaction has been confirmed and discussed in detail below (Figure S4). In the case on SG and Au.01 catalysts we see a gradual decrease in the conversion towards $NO_2$, however the activity of this materials is maintained, and a steady state has been reached.

Photocatalytic activity of studied catalysts was assessed by evaluation of photocatalytic NO conversion rate (PR) calculated in respect to $NO_2$ formation and $NO_x$ removal. Photocatalytic rate PR ($\mu mol\ m^{-2}\ s^{-1}$) was calculated according to the formula:

$$PR = \frac{C_r}{A} \times V \qquad (2)$$

where:

$C_r$—$C_{NO2out}$-$C_{NO2in}$ or $C_{NOxin}$-$C_{Noxout}$ ($\mu mol\ m^{-3}$), A—photocatalyst surface ($m^2$), V—reaction gases flow rate ($m^3\ s^{-1}$). This parameter was calculated both for maximum conversion of NO to $NO_2$ at the beginning of irradiation and for steady state concentrations (Figure 4).

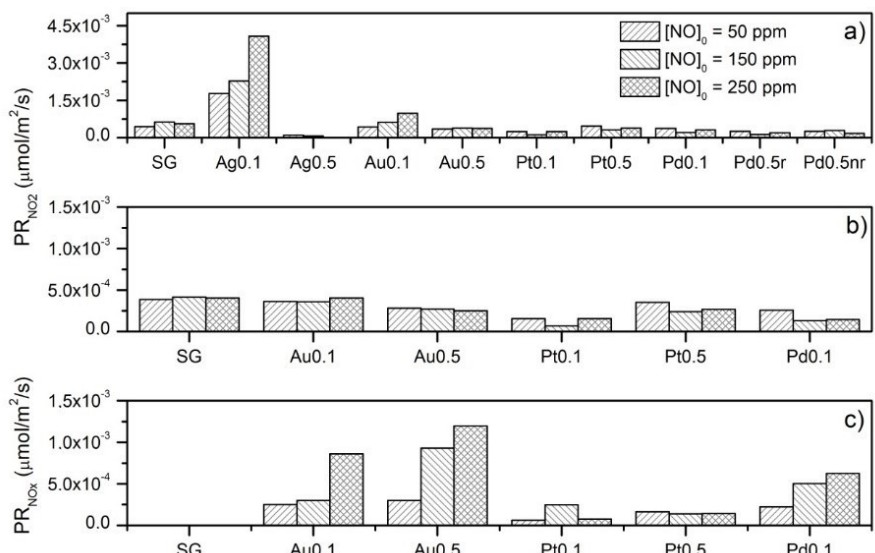

**Figure 4.** Photocatalytic rate of oxidation NO into $NO_2$ ((**a**)—maximum PR, (**b**)—steady state PR), (**c**)—photocatalytic rate of $NO_x$ removal.

$PR_{max}$ is presented in Figure 4a for all the studied materials. The highest PR values were obtained for Ag0.1. Almost four times lower values were obtained for SG and Au0.1. Figure 4b shows only 6 synthesized materials, this is because no steady state was reached for Ag0.1, Ag0.5, Pd0.5r, Pd0.5nr catalysts. It can be observed that similar $PR_{ss}$ values were recorded for the process carried out over SG catalysts and Au0.1, whereas lower values were obtained for the rest of the materials. Hence, the activity of SG and Au0.1 towards $NO_2$ formation is almost equal. The influence of noble metals is clearly visible in Figure 4c. The highest $NO_x$ removal rate was obtained over gold decorated $TiO_2$, i.e., Au0.1 and Au0.5 and palladium decorated catalyst Pd0.1. The best results were obtained over Au0.5 catalysts, showing not only higher $NO_x$ removal but also good selectivity towards $NO_2$. It has been shown that pure $TiO_2$ (SG) catalyst acts mainly as the NO converter into $NO_2$. In the case of noble metal decorated materials the reaction of photocatalytic oxidation seems to proceed further. It turns out that also for smaller initial NO concentrations (500 ppb) the higher photocatalytic activity is also observed for the gold decorated $TiO_2$ [15]. It is important to note here that Hernández Rodríguez et al. (2017) have used P25 based gold and platinum decorated catalysts.

In order to determine the fate of NO over metal decorated catalyst, surface species present on the catalyst surface were also analyzed by HATR/FTIR before and after photocatalytic reaction (Figure S4). The spectrum obtained for $TiO_2$ film before the reaction was depicted in black. Two characteristic peaks can be observed at around 3400 and 1640 $cm^{-1}$. First broad peak was attributed to the stretching vibrations in OH surface groups or physically bonded water. The second peak was connected with the scissor vibrations in O-H-O. These two peaks confirmed the presence of adsorbed water at the surface of immobilized films before the photocatalytic process. The second spectrum depicted with blue was taken after photocatalytic oxidation of nitrogen oxide. First, the presence of an additional doublet peak around 1300 $cm^{-1}$ can be noticed. Whereas the intensity of the remaining peaks has significantly increased when compared with the spectra obtained before reaction. 1317 $cm^{-1}$ peak was recognized as asymmetric stretching vibrations of O-N-O, confirming the presence of nitric acid at the immobilized catalyst. The second peak at 1390 $cm^{-1}$ also confirms $HNO_3$. It is consistent with the presence of bending vibrations in H-O-N in the nitric acid molecule [19]. The wide band between 2800 and 3750 $cm^{-1}$ was in this case connected both with the presence of the adsorbed water and stretching vibrations O-H in $HNO_3$ molecule [19]. Therefore, it has been confirmed that during photocatalytic oxidation of nitrogen oxide, reaction products namely $HNO_3$ are adsorbed at the photocatalyst surface. This also confirmed that the $NO_x$ removal is a result of NO photocatalytic oxidation and not a reduction.

Based on the obtained results we concluded that the enhanced photocatalytic activity of Au0.5 catalyst in comparison to SG was caused by modification of $TiO_2$ surface with gold nanoparticles. Two mechanisms can play a role in the enhanced activity metal-semiconductor junction formation between $TiO_2$ and gold nanoparticle under UV light and surface plasmon resonance effect in visible light [8,12] (Figure 5).

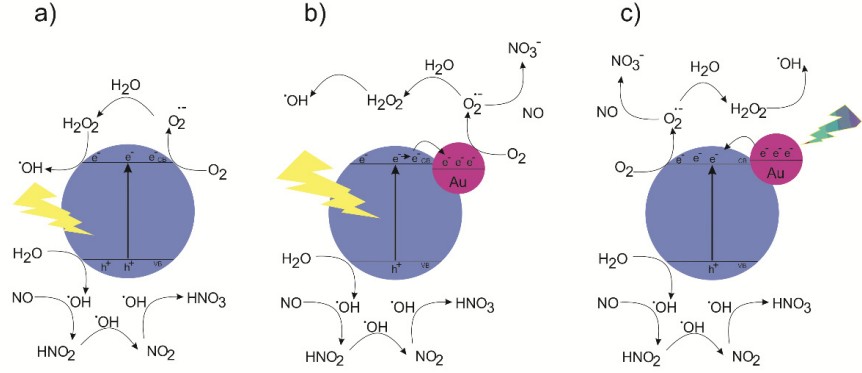

**Figure 5.** Schematic representation of mechanism of NO photocatalytic oxidation over SG (**a**) and Au0.5 photocatalyst under UV (**b**) and visible light (**c**).

Pure $TiO_2$ (SG) (Figure 5a) in the form of anatase has limited photocatalytic activity due to the fast charge recombination. The formation of hydroxyl radicals is too slow to convert first NO into $NO_2$ and then to $HNO_3$. In the case of Au0.5 material nanoparticles of gold on the surface of $TiO_2$ work as electron traps. This lowers the rate of electron-hole recombination. Photogenerated electrons that have moved to the gold nanoparticle can more easily react with electron acceptors ($O_2$) (Figure 5b). This is due to the high work function of gold (5.4–5.75 eV) [20]. Existence of gold nanoparticles is further beneficial for the absorption of visible light. SPR promotes excitation of electrons from Au nanoparticles, that transfer to conduction bad (CB) of $TiO_2$ (Figure 5c) improving charge separation.

This leads to higher hydroxyl radical generation and at the same time formation of $NO_3^-$ by superoxide anion and nitric oxide. In our opinion the increased activity of the gold decorated $TiO_2$, in comparison with other noble metals, might be a result of chemical resistance of gold. This is especially important in the presence of high initial NO concentrations and NO photocatalytic oxidation final product (i.e., $HNO_3$), which is a strong

oxidant, since deactivation of other materials has been observed. The behavior was clearly visible for Ag0.1 material (deactivation of the catalyst within 60 min of irradiation). Higher activity of gold decorated material is therefore, on one hand a result of metal-semiconductor junction that is formed and on the other hand a consequence of gold chemical resistance. The final reaction product is adsorbed at the surface of photocatalyst but can be easily removed by water or steam regeneration.

## 3. Materials and Methods

### 3.1. Materials

Nitrogen monoxide was supplied from a gas cylinder (0.5% vol. NO in $N_2$) (AIR PRODUCTS). Titanium (IV) isopropoxide (TIP) (97%) was purchased from Aldrich Chem. and used as titanium source for the preparation of $TiO_2$ nanoparticles. $HAuCl_4$ (Au~52%), $PdCl_2$ (5% wt. solution in 10% wt. HCl), $AgNO_3$ (99%) and $K_2PtCl_4$ (99.9%) (Sigma-Aldrich) were used as metal sources in the preparation procedure. Sodium borohydride ($NaBH_4$, 99%) (POCh S.A.) was used as the reducing agent. Additionally, sulfuric acid (95%) (CHEMPUR) was used as well as sodium salicylate, sodium hydroxide and potassium sodium tartrate 4 hydrate (POCH S.A.).

### 3.2. Synthesis of Photocatalysts

Noble metals decorated catalysts were synthesized with the use of the sol-gel method. Hydrolysis reaction of titanium isopropoxide with water was used. 25 $cm^3$ of TIP was mixed with ethanol and deionized water in volumetric ratios ($V_{TIP}/V_{ROH} = 1$; $V_{TIP}/V_{H2O} = 1.75$). Water addition was kept at a rate of 1 $cm^3$ $min^{-1}$. The thick precipitate was formed which gradually peptized for 2 h to form a clear sol. Finally, a certain amount of metal precursor ($AgNO_3$, $PdCl_2$, $HAuCl_4$, and $K_2PtCl_4$) was dissolved in the deionized water at room temperature and mixed with $TiO_2$ gel. The metal precursor was introduced into the solution in amounts: 0.1, 0.5% mol. Ag; 0.1, 0.5% mol. Au; 0.1, 0.5% mol. Pt; 0.1, 0.5% mol. Pd, related to $TiO_2$. Additionally, for one sample Pd 0.5% mol.-$TiO_2$ the reducer $NaBH_4$ was used. The molar ratio of the reducing agent to metal ions was equal to 3. The prepared solutions were mixed for 1 h at ambient temperature and afterwards the obtained gel was dried for 24 h at T = 80 °C. The final step of the powder preparation was its calcination for 2 h at 400 °C. Previously, it was shown that this calcination temperature had provided the highest photoactivity of the synthesized materials [21]. Detailed data for the synthesized photocatalysts were presented in Table S1.

### 3.3. Analytical Methods

The crystalline structure and crystallite size in all the samples was examined by XRD (X-ray diffraction) in Pert Pro-MPD (Philips) diffractometer with copper anode ceramic roentgen lamp ($\lambda$ = 1.542 Å). The XRD pattern was measured in the 2θ range 20–65°.

The nitrogen adsorption-desorption isotherms were recorded at the liquid nitrogen temperature (77K) on a Micromeritics Gemini V (model 2365) and the specific surface areas were determined by the Brunauer–Emmett–Teller (BET) method in the relative pressure ($p/p_0$) range of 0.05–0.3. All the samples (mass 0.4 g) were degassed at 200 °C for 2 h prior to the nitrogen adsorption measurements. From the obtained nitrogen adsorption isotherm, the BET surface area was calculated according to Brunauer, Emmet and Taller equation. Micrometrics Gemini V unit enabled a pore size determination in the studied nanopowders.

The surface atomic composition was analyzed with the use of XPS (X-ray Photoelectron Spectroscopy) Kratos AXIS Ultra spectrometer with a monochromatic Al K$\alpha$ X-Rays source (excitation energy equal to 1486.6 eV). An analysis area of 300 × 700 μm was used for all measurements. The anode power was set at 150 W and the hemispherical electron energy analyzer was operated at pass energy 20 eV for all the high-resolution measurements. All measurements were performed with the use of a charge neutralizer set individually for each analytic position on samples' surface. Evaluation of XPS data was conducted by means of a Kratos Vision 2 software. The background subtraction was performed with a

Shirley algorithm and the adventitious carbon main peak (C 1s, 284.8 eV) was used for a final calibration of each spectrum.

The morphology and size characterization of metal nanoparticles deposited on $TiO_2$ were carried out with Cs-corrected STEM (High Angle Annular Dark Field, HAADF). For the sample preparation a drop of M-$TiO_2$ aqueous dispersion was deposited onto a copper grid covered with a formvar-carbon membrane. All the samples were imaged with a Jeol model ARM 200F operating at 200 kV with an EDX analyzer (JED2300). STEM images were analyzed using ImageJ, using the particle analysis feature. Additionally, for gold sample energy dispersive X-ray spectroscopy was used to evaluate gold nanoparticles distribution on $TiO_2$ material (EDX Oxford INCA 250 spectrometer). Measurements by EDX technique were performed at least at 17 different spots for a given sample, and then average atomic concentration and its standard deviation were calculated for identified elements. Electron energy of 25 eV was used and the $400 \times 400$ µm area was scanned each time. All measurements were performed in nitrogen gas atmosphere (150 Pa).

*3.4. Photoactivity Study*

The photocatalyst immobilization on glass plates (11 cm$^2$) was carried out by an even distribution of synthesized catalysts suspension in deionized water. The suspension was treated for 30 min with ultrasounds to minimalize powder agglomeration. So prepared immobilized photocatalysts were dried in 120 °C for 20 h.

The operating set-up was presented in Figure 6. The volume of the photocatalytic reactor was equal to 46 cm$^3$. Glass plates with immobilized $TiO_2$ films were placed in the reactor and irradiated with the use of a xenon lamp 450 W (light intensity UV-Vis $\approx$ 110 W m$^{-2}$).

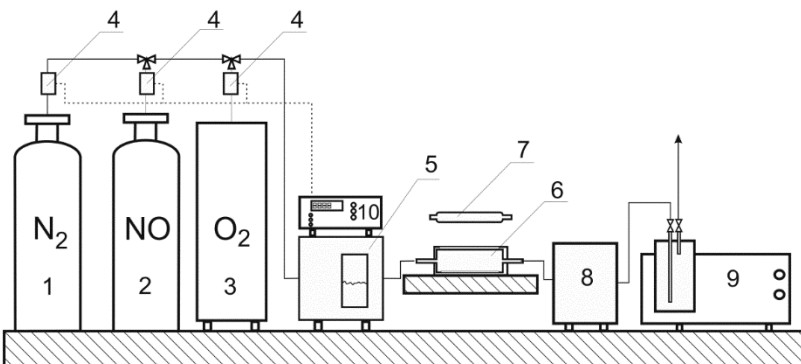

**Figure 6.** Experimental set-up: 1—N$_2$ gas cylinder, 2—NO in N$_2$ gas cylinder C = 0.5%—AIR PRODUCTS, 3—oxygen generator PSA—AirSep Onyx, 4—mass flowmeters—Brooks Instruments SLA5850, 5—humidifying unit—LAT UNG-1, 6—photocatalytic reactor, 7—450 W xenon lamp—Osram, 8—drying unit—LAT OS-3, 9—FTIR spectrometer—Jasco FTIR 4200 equipped with the gas cuvette, 10—control panel for mass flow meters—Brooks Instruments.

Reaction gases flow rate was controlled by mass flowmeters. The total flow rate of reaction gases was set to 1.5 dm$^3$ min$^{-1}$ (residence time—1.8 s). The mixture of reaction gases consisted of nitrogen monoxide (initial concentrations: 50–250 ppm), oxygen (around 20% vol.) and nitrogen. Prior to entering the photocatalytic reactor, the mixture was passed through a humidifying unit ($X_{H2O} \approx 2.7$ g kg$^{-1}$). The gas stream containing NO was then irradiated in the reactor (T $\approx$ 25 °C). Gases after irradiation were introduced into a drying unit to eliminate water vapor and finally were analyzed by an FTIR spectrometry. During the FTIR measurements in a gas cuvette (volume: 0.1 dm$^3$, optical path: 2.4 m) the spectra from 700 cm$^{-1}$ to 4000 cm$^{-1}$ were monitored with 4 cm$^{-1}$ resolution. Each spectrum was an average of 50 scans.

Final reaction products adsorbed at the photocatalyst surface were identified with the use of HATR/FTIR. For selected experiments glass plates with immobilized $TiO_2$ were analyzed before and after the photocatalytic oxidation of NO. Jasco FTIR 4200 was

equipped with the HATR (Horizontal Attenuated Total Reflectance) unit with ZnSe crystal (45° degrees face angle) by PIKE Technologies.

## 4. Conclusions

The $TiO_2$ based photocatalysts decorated with four noble metals have been investigated in this work. The photocatalytic oxidation of nitrogen oxides proceeds with the initial formation of $NO_2$ over SG and noble metal decorated catalysts Au0.1, Au0.5, Pt0.1, Pt0.5, Pd0.1. However, $NO_x$ removal was observed only with the modified catalysts. The $NO_x$ removal increased in following order Au0.5 > Au0.1 > Pd0.1 > Pd0.1 ≈ Pt0.1. We showed that gold is the most promising noble metal for this application. Additionally, we have confirmed that during photocatalytic oxidation of nitrogen oxide, reaction products, namely $HNO_3$, are adsorbed at the photocatalyst surface.

**Supplementary Materials:** The following supporting information can be downloaded at: https://www.mdpi.com/article/10.3390/catal12080857/s1, Figure S1: XRD patterns of decorated-PCs, Figure S2: EDS components analysis for Au0.5 photocatalyst, Figure S3: NO conversions into $NO_2$ profile (a) SG, (b) Ag0.1 (c) Au0.1, Figure S4: HATR/FTIR spectra obtained for photocatalyst film before and after photocatalytic reaction; Table S1: Photocatalysts name, precursor used and its concentration together with PCs colors; Table S2: Basic characteristics of studied photocatalysts.

**Author Contributions:** Conceptualization: K.S.; investigation: K.S., A.M., J.B., M.G. and G.N.; resources: K.S., S.J. and A.Z.-M.; writing—original draft preparation: K.S., A.M. and J.B.; writing—review and editing K.S., A.M., J.B., M.G., G.N., S.J. and A.Z-M.; project administration: K.S.; supervision K.S. and A.Z.-M., funding acquisition: K.S. and S.J. All authors have read and agreed to the published version of the manuscript.

**Funding:** This research was funded by Polish Ministry of Science and Higher Education grant number IP2011 049871 and National Centre for Research and Development, Poland grant number PBS1/A9/13/2012.

**Data Availability Statement:** Data is contained within the article.

**Acknowledgments:** KS thanks to Jan Sielski from the Molecular Engineering Department, Lodz University of Technology for performing EDX analysis.

**Conflicts of Interest:** The authors declare no conflict of interest.

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
