# Peer review of "NOx Photooxidation over Different Noble Metals Modified TiO2"

_catalysts, doi:10.3390/catal12080857_

Round 1

Reviewer 1 Report

This research paper entitled “NOx photooxidation over different noble metals modified TiO2” demonstrated a good photocatalytic activity of noble metal modified TiO2 to remove NO from air via catalytic nitrogen oxide oxidation process.

The paper is well organized and easy to read. As a fundamental research work on nanomaterials, this paper can be accepted to be published in Catalysts after minor revision.

However, there are some fundamental problems that the authors should take in account in your research work:

11)      TiO2 has been suspected of being carcinogenic by the American Food and Drug Association[i],[ii].So, use the TiO2 based photocatalysts can be harmful.

22)      TiO2 used in this work was in free powers just immobilized on glass surface. This method is not sustainable. Furthermore, the nanoparticle manipulation could be dangerous for researchers and worker in future application.

Thus, the proposed method is not a promising & sustainable way to treat air purification. But as a fundamental research paper, the results are interesting.

Other points:

11)      The DRS UV-vis method has not been presented in paragraph 2.3.

22)      Please harmonize all characters’ size.

33)      Add a unit to "2 Theta" in Figure S1.

44)      Line 113: λ = 1,542 Å ==> λ = 1.542 Å.

55)      All “â—¦C” ==> °C and without space between number & °C.

[i] S. Bettini, E. Boutet-Robinet, C. Cartier, C. Coméra, E. Gaultier, J. Dupuy, N. Naud, S. Taché, P. Grysan, S. Reguer, N. Thieriet, M. Réfrégiers, D. Thiaudière, J.-P. Cravedi, M. Carrière, J.-N. Audiot, F. H. Pierre, L. Guyzlack-Piriou, E. Houdeau, Sci. Rep. 2017, 7, 40377.

[ii] IARC Working Group, IARC Monographs on the Evaluation of Carcinogenic Risks to Humans, Vol. 93, World Health Organisation 2010.

Author Response

Thank you for the reviewer’s report, we have tried to answer your questions as follows:

  1. The authors thank for the comments regarding TiO2 and its applicability in the future based on the recent research and regulations. The introduction (lines 65-68) has been modified to incorporate this comment.

  1. The authors agree that the the method used in this research for immobilization is not the method that has potential for industrial applications. In these studies, the aim was to assess the influence of noble metals in the system on the efficiency of TiO2.

other points have been corrected in the text and highlighted in yellow.

Reviewer 2 Report

The manuscript discusses the synthesis and characterization of TiO2 nanoparticles decorated with different noble metals for the abatement of NOx. The work contains results which can be of interest for the readers of Catalysts, however I think that several aspects should be clarified and improved before publication. I will report in the following my issues about the manuscript:

1  1) The chemical composition of the samples obtained by XPS analysis (Table 1) is not consistent with the 2:1 atomic ratio between O and Ti expected for TiO2, although the reported errors are low. The authors should clarify the reasons for these deviations.

2) The samples contain a high fraction of surface carbon (about 20 %at), as highlighted by XPS, likely coming from the precursors employed during the synthesis which are not completely burnt during the calcination process which was performed at a rather low temperature (i.e. 400 °C), Which is the impact of these abundant carbon-containing groups on the functional properties and photocatalytic activity of the materials?

    3) A more detailed analysis of the XRD patterns should be performed, analyzing the variation of the average crystal size in the different crystallographic directions of the samples considering the FWHM of the XRD peaks (using Scherrer analysis or similar methods, see e.g. ACS Appl. Nano Mater. 2018, 1, 5355−5365). These average domain size obtained by XRD analysis could be also compared with the electron microscopy dimensional data and BET results to have a more complete characterization of the TiO2 particles size and morphology.

4) In  the introduction the authors correctly say that “Ag, Au, Pt or Pd nanoparticles can promote both activity under UV and Vis light, through restraining electron-hole recombination or appearance of surface plasmon resonance (SPR)”. However, for their photocatalytic tests they employed a xenon lamp, which emits both UV and visible light. Therefore, it is not possible to understand if the observed increase in photocatalytic activity after doping is due to a plasmonic effect, or to an increased charge separation, or to both effects together. To clearly prove if also a plasmonic effect is present, the authors should perform additional tests comparing bare TiO2 and metal decorated TiO2 using pure visible light.

 5) The authors hypothesize that the higher photoactivity of Au-decorated TiO2 is due to a higher resistance of gold to the strongly oxidant environment. This speculation should be proven showing some experimental evidence for the reason of the deactivation of the samples containing other metals (e.g. changes of oxidation state by XPS).

 6) The number of reported significant figures is not always consistent with the corresponding experimental uncertainty which should contain only 1 or 2 figures (see e.g. Table 1).

 7)      A specific “Conclusions” section is missing, and I think that it should be added after section 3.2, summarizing the main findings of the study.

Author Response

Thank you for the reviewer’s report, we have tried to answer your questions as follows. All the changes made to the text have also been highlighted in yellow:

  1. The O2/Ti ratio is between 2.3and 2.5. This is caused by the extra oxygen that is bonded with carbon also detected in the samples. Please see lines 198 and 199 “In the oxygen O 1s spectrum two components are present (Fig. 2b) at 529.7 eV and 531.3 eV. They were assigned to TiO2 and oxygen in hydroxyl or carbonyl groups bonded with an adventitious carbon, respectively”.

  1. Reviewer is correct in suggesting the source of carbon in our samples. The role of carbon species can be to promote activity in visible light (Huang et al., 2012). However, in this work the carbon content on the photocatalytic activity as carbon content in all samples was similar, and therefore did not confound the results.

  1. According to reviewers’ suggestion the Scherrer analysis has been performed, and the results have been included in the text of the manuscript (lines 181-185). Below, please see a table with detailed values.

Crystallite size (nm)

(101)

(004)

(200)

Au0.1

9

6

8

Au0.5

9

8

8

Ag0.1

9

6

8

Ag0.5

8

5.5

7.5

Pd0.1

10

6

10

Pd0.5nr

9

5

8

Pd0.5r

9

4.5

7

Pt0.1

9

6

8

Pt0.5

8

5

7.5

  1. Based on reviewer’s suggestion and the lack of opportunity to perform new experiments with the use of only visible light, we corrected the part concerning mechanism (lines 334-336) and corrected both Figure 5 and graphical abstract to include both possible mechanisms.

  1. This has been clarified in the text (lines 270-271) . The catalysts that were inactivated were those containing silver and palladium (no steady state was obtained for them). In respect to silver modified catalysts we additionally observed clear discoloring of the catalysts surface during the reaction. We did not test the XPS after reaction, as the catalysts would need to be firstly scraped from the glass plates and hence the XPS analysis (which shows only surface species) would not provide a clear answer to the oxidation state of metals. However, the adsorption of HNO3 at the surface of catalysts has been confirmed by ATR-FTIR as well as colorimetric analysis based on the reaction of nitrates with sodium salicylate in the presence of sulfuric acid, performed on DI water used for post-treatment rinsing of the catalyst (results not included in this manuscript). The deactivation happens as a result of both coverage with reaction product HNO3 and its surface reactions with catalysts components, e.g. silver.

  1. The numbers of significant figures have been corrected.

  1. The conclusion part has been added, according to the suggestion.

Huang C-H, Lin Y-M., Wang I-K., Lu Ch-M., Photocatalytic Activity and Characterization of Carbon-Modified Titania for Visible-Light-Active Photodegradation of Nitrogen Oxides, International Journal of Photoenergy, 548647, 2012.

Reviewer 3 Report

The authors have synthesized noble metal decorated TiO2 for the photocatalysis of NOx gasses. They have presented a novel method of determining the reaction by-products through gas phased FTIR. I feel the content of this manuscript is novel and would be useful to researchers working on environmental catalysis and recommend publishing the manuscript as is.

Author Response

Thank you for the reviewer’s report.

Round 2

Reviewer 2 Report

The authors replied properly to all my issues, except comments 4 and 5. In particular:

1)    I do not agree with the modification of Figure 5 since the authors did not perform the requested experiments under visible light. Therefore, the possible surface plasmon resonance effect proposed in Figure 5c is a mere speculation and cannot be demonstrated by the experimental results as stated by the authors (page 7, lines 232-235). This issue should be better discussed in the text.

2) The authors should add in the manuscript the experimental data mentioned in their reply showing that the deactivation happens also as a result of surface coverage by the reaction product HNO3 to support the new sentence added on page 6, lines 181-182.

Author Response

Thank you for your comments, please find our responses below:

  1. We understand the concern of the reviewer, hence the manuscript text has been corrected to clarify that both those mechanisms are possible, but their relative importance has not been a subject of this study (lines 235-236).

  1. We apologize for the confusion, resulting from our previous answer. We have clarified this in the text on lines 182-183. The HATR-FTIR data confirming presence of nitrates at the surface of catalysts after reaction have been already shown in the supplementary materials file in figure S4 and were in detail discussed in lines 215-233. What is not shown, are the results of colorimetric analysis. They showed that TiO2 plate wash contained nitrates after photocatalysis reaction. The obtained values were in the range 0.0017 to 0.016 mg NO3- per TiO2 plate depending on the initial NO concentration. As the HATR-FTIR spectra confirm already the presence of HNO3, the colorimetric method description and results have not been added to the text.